# Scoping Review of Studies on Affective–Psychological and Social Characteristics of South Korean Engineering Students

**DOI:** 10.3390/bs15091189

**Published:** 2025-08-30

**Authors:** Soonhee Hwang

**Affiliations:** Department of Liberal Arts & Science, Hongik University, 2639 Sejong-ro, Jochiwon-eup, Sejong-si 30016, Republic of Korea; soonheehwang@hongik.ac.kr

**Keywords:** engineering students, learner characteristics, affective–psychological characteristics, social traits, scoping review, self-efficacy, engineering education

## Abstract

This scoping review examines the affective–psychological and social characteristics of undergraduate engineering students in South Korea, identifying key research trends, thematic focuses, and gaps in the literature. A total of 95 peer-reviewed articles published between 2000 and 2024 were analyzed based on publication year, journal outlet, research topics, and related variables. The literature search was conducted using major databases, including RISS, KCI, and DBpia. The findings highlight self-efficacy—particularly domain-specific self-efficacy—as a core construct linked to academic achievement, persistence, and career development. Social competencies such as communication, teamwork, and convergence ability are also emphasized; however, limited attention has been paid to emotional resilience, burnout, and ethical responsibility. Despite their growing importance in the artificial intelligence-driven era, gender differences, digital literacy, and global competencies remain underexplored. These findings underscore the need for learner-centered, evidence-based instructional strategies, as well as more longitudinal, comparative, and intervention-focused studies. This review offers foundational insights for designing inclusive, future-oriented educational programs tailored to the diverse needs of South Korean undergraduate engineering students.

## 1. Introduction

Science, technology, engineering, and mathematics in higher education is globally recognized as a strategic pathway for cultivating future industry leaders. In response to the growing demands of artificial intelligence (AI), digital transformation, and sustainable innovation, many countries have restructured their curricula and support systems to develop relevant competencies ([5]; [111]). In South Korea, rapid technological advancement—particularly in AI—has underscored the critical role of engineering students in national development ([48]). Consequently, understanding the affective–psychological and social characteristics of engineering majors is essential for informing effective educational policies and practices. Engineering education in South Korea has undergone substantial reform in response to the Fourth Industrial Revolution, with increased emphasis on digital literacy and future-oriented skills. Parallel to these changes, research has expanded to examine factors influencing engineering students’ academic performance, persistence, and career trajectories in South Korea ([41], [42], [46]). South Korea’s unique educational landscape—characterized by intense academic competition, rigid university admissions processes, and institutional disparities—exacerbates stress and significantly shapes the psychological and social development of students. Engineering, as a discipline, emphasizes logical reasoning and problem-solving. However, within South Korea’s high-pressure academic environment, engineering students often face mental health challenges linked to academic stress and uncertainty as documented in Korean studies ([42], [46]). These challenges, in turn, influence their learning engagement and strategies ([100]). Learner characteristics such as structured thinking, self-efficacy, and resilience are especially relevant in this context and are frequently explored in relation to student outcomes ([15]; [93]).

Affective–psychological and social characteristics play a critical role in curriculum design and learner support systems ([48]; [55]; [58]). Traits such as self-efficacy, emotion regulation, collaboration, and empathy are widely recognized in both educational psychology and engineering education as essential to student success ([11]; [143]). Moreover, the increasing integration of AI technologies in education has been shown to influence students’ internal and social attributes, including motivation, self-regulation, and empathy ([129]). Despite growing interest in these areas, research in engineering education continues to prioritize cognitive and technical competencies, often neglecting values, attitudes, and ethical development ([28]; [42]). However, constructs such as identity, emotional well-being, and affective engagement have been shown to significantly influence student motivation and persistence ([84]; [106]; [137]). Noncognitive competencies—such as teamwork, communication, and grit—are also increasingly recognized as vital for success in engineering careers ([31]; [27]; [121], [122]; [136]). Nevertheless, much of the existing research has examined these characteristics in isolation, employed inconsistent definitions, and lacked comprehensive theoretical or integrative frameworks.

Against this backdrop, the present scoping review aims to examine academic literature on the affective–psychological and social characteristics of undergraduate engineering students in South Korea. Scoping reviews are particularly well suited for mapping broad and conceptually diffuse research landscapes, clarifying terminology, and identifying emerging trends and research gaps ([6]; [27]; [120]). Unlike systematic reviews, which emphasize study quality and evidence synthesis, scoping reviews are designed to provide a comprehensive overview of the existing literature, making them especially appropriate for underexplored or conceptually fragmented fields ([6]; [120]). In engineering education, recent scoping reviews have addressed a range of affective–psychological and social domains and instructional approaches aligned with the present study’s focus. These include investigations of constructs related to student thriving—such as self-efficacy, motivation, and social connectedness ([27]); mental health interventions, including mindfulness and mentoring programs ([137]); emotion-centered research, which has predominantly focused on anxiety while overlooking affective diversity and identity development ([106]). Previous reviews have also synthesized belief-related constructs such as self-efficacy and mindset ([84]), examined curriculum-level reforms in sustainability education ([28]), and explored engineering practices under the Next Generation Science Standards framework ([12]; [23]; [101]). Additional work has categorized instructional contextualization ([82]), investigated sociotechnical thinking ([125]), and highlighted the role of self-regulated learning strategies ([139]). Furthermore, several scoping reviews have assessed the application of systematic review methodologies in engineering and engineering education, raising concerns about methodological rigor and adherence to established guidelines such as Preferred Reporting Items for Systematic Reviews and Meta-Analyses (PRISMA) ([124]; [121], [122]).

The present scoping review aims to systematically map the literature on the affective–psychological and social characteristics of South Korean undergraduate engineering students. It is guided by the following research questions (RQs):

RQ1. What are the publication trends (e.g., years, journals) in empirical studies on the affective–psychological and social characteristics of engineering undergraduates in South Korea?

RQ2. What are the major research themes and core constructs explored in this body of work?

RQ3. What individual or contextual factors have been identified as influencing or associated with these characteristics?

By synthesizing this literature, the review seeks to inform educators, researchers, and policymakers on how to better support the holistic development of engineering students and to identify promising directions for future research.

## 2. Literature Review

### 2.1. Learner Characteristics

Learner characteristics are commonly categorized into cognitive, affective–psychological, and social domains ([55]; [109]; [128]). These domains are interrelated and collectively influence motivation, engagement, and learning strategies ([126]). For instance, self-regulated learning not only enhances academic motivation but also strengthens self-efficacy ([129]). While cognitive traits underpin reasoning and problem-solving abilities, affective–psychological and social factors—such as self-efficacy and interpersonal dynamics—play a critical role in collaboration and overall academic performance ([11]).

### 2.2. Affective–Psychological Characteristics

Affective–psychological characteristics refer to learners’ internal states related to emotions, motivation, and self-concept. These traits are closely linked to academic achievement, persistence, and problem-solving ability. Key constructs within this domain include self-efficacy, motivation, personality, psychological well-being, thinking styles, grit, stress, and emotional regulation. Self-efficacy—defined as an individual’s belief in their ability to complete tasks—strongly predicts motivation, academic performance, and learning behavior ([11]). Motivation, both intrinsic and extrinsic, plays a central role in initiating and sustaining learning ([126]). Personality, as an affective factor, influences learners’ preferences in ability, needs, interests, and motivation, thereby shaping learning styles and academic outcomes, as shown in studies of South Korean engineering students ([93]). Psychological well-being, a multidimensional construct reflecting an individual’s optimal functioning and the realization of potential, represents a healthy and flourishing mental state ([127]). Among university students, psychological well-being significantly influences academic adjustment, career planning, emotional regulation, interpersonal relationships, and satisfaction with one’s major ([40], [41]). Thinking styles—positioned at the intersection of cognition and personality—describe learners’ preferred ways of applying their abilities. These styles influence creativity, teamwork, and communication, and are critical in collaborative academic settings ([37]; [49]; [135]). Grit—defined as sustained passion and perseverance toward long-term goals—is positively correlated with conscientiousness and has been shown to predict academic achievement through persistent effort ([26]; [95]). Successful adaptation to university life requires a combination of academic competence, emotional stability, and interpersonal skills, all of which are closely linked to overall adjustment ([10]; [58]). In engineering education, psychological factors such as stress, depression, and anxiety play a central role in students’ academic adjustment and career decision-making. Emotional resilience and the ability to regulate stress are also significant predictors of academic success ([123]). Emotional regulation strategies—defined as the ability to manage emotional responses in socially and culturally appropriate ways under stress—enable effective problem-solving and adaptation ([17]). Successful emotional regulation supports students’ psychological well-being, enhances interpersonal relationships, and improves conflict resolution. It is also closely associated with academic stress management and sustained learning engagement. Positive academic emotions can enhance motivation and achievement, whereas negative emotions often impede cognitive processes and hinder learning outcomes ([119]).

### 2.3. Social Characteristics

Social characteristics refer to interpersonal competencies such as communication, collaboration, empathy, leadership, and social responsibility ([51]). These traits are essential in team-based learning environments and are increasingly recognized as critical for academic and professional success in a digital, globalized world. Core competencies are defined as essential skills individuals must develop throughout life to effectively navigate complex and evolving social demands ([59]; [110]). Accordingly, researchers and international organizations have identified communication, self-directedness, collaboration, Information and Communication Technology proficiency, and problem-solving as foundational lifelong competencies ([59]). These competencies—rooted in social abilities such as teamwork, communication, and collaborative problem-solving—are strongly associated with both academic performance and future employability. Empathy, encompassing both cognitive and affective dimensions, is also crucial for understanding others’ perspectives and responding appropriately ([30]), with evidence from South Korean engineering undergraduates ([44]). It supports the development of integrative competencies needed in interdisciplinary and collaborative contexts. In engineering education specifically, teamwork competence—alongside creativity—is considered a core skill for effective practice and innovation ([32]). Engineering tasks and projects are often carried out collaboratively, with teamwork and creativity playing critical roles in fostering innovation and effective problem-solving ([7]). The ability to function effectively in team settings is influenced by multiple factors, including organizational and team dynamics, individual thinking styles, personality traits, cognitive abilities, creativity, and problem-solving skills ([8]; [50]). Self-leadership—defined as self-motivation and goal-setting—was initially explored within the context of organizational innovation ([107]) and has since been linked to career adaptability, self-efficacy, academic satisfaction, and a range of academic competencies ([43]). Moreover, the rapid advancement of science and technology has introduced complex societal challenges, highlighting the growing importance of social responsibility in scientific and engineering professions. Social responsibility extends beyond individual ethical behavior (e.g., honesty, integrity) to include collective efforts aimed at advancing the well-being of society, humanity, and the environment ([83]).

### 2.4. Integrated Characteristics

Affective–psychological and social characteristics are deeply interconnected, jointly influencing student behavior, engagement, and academic outcomes ([11]; [143]). For instance, effective emotion regulation can enhance collaboration and reduce the negative impact of anxiety on motivation and learning engagement ([123]; [51]). Drawing on self-regulated learning theory, [129] ([129]) demonstrated how affective–psychological traits, in interaction with environmental factors, shape learners’ motivation and self-regulation—both of which are critical for academic achievement.

Building on this framework, the present study categorizes research that simultaneously addresses affective–psychological and social characteristics as integrated trait studies. It further seeks to explore how these two domains are linked and co-constructed within the context of engineering education.

## 3. Study Methodologies

### 3.1. Search Process and Selection Criteria

This study employed a scoping review to systematically analyze research on the affective–psychological and social characteristics of undergraduate engineering students in South Korea, with the aim of identifying the educational contributions and limitations of existing studies. A scoping review is a form of knowledge synthesis that maps key concepts, research types, and gaps within a defined field. Unlike systematic reviews, scoping reviews typically do not assess the quality of included studies but are particularly useful for exploring complex or heterogeneous bodies of literature ([6]; [102]).

The review was conducted in accordance with the PRISMA 2020 guidelines, which are widely adopted for comprehensive and transparent syntheses of diverse research topics ([112]). In addition, this review adhered to the PRISMA-ScR extension for scoping reviews ([138]); the completed PRISMA-ScR checklist is provided in Appendix A. The review process consisted of the following five stages:Establishing inclusion and exclusion criteria;Identifying and retrieving relevant literature from major academic databases;Screening records through title, abstract, and full-text review;Systematically analyzing and synthesizing key findings from the selected studies;Presenting results with a focus on educational implications and research limitations.

### 3.2. Literature Search and Selection

To conduct a comprehensive scoping review of research on the characteristics of engineering students in South Korea, three major academic databases were searched: the Research Information Sharing Service, the Korea Citation Index, and DBpia. The search targeted peer-reviewed journal articles published between 2000 and 15 October 2024.

A broad and inclusive search strategy was employed to ensure comprehensive coverage and reduce the risk of omitting relevant studies. This strategy utilized compound keyword combinations and variant terms, including “engineering college” + “university students” or “undergraduates,” “college of engineering students,” “engineering undergraduates,” “engineering majors,” and “students in engineering disciplines.” To further expand the scope, English-language terms such as “engineering students” and “engineering + students” were also included. Only articles published in peer-reviewed journals indexed in the Korea Research Foundation’s list of registered or candidate journals were considered, in order to maintain academic rigor and quality. The literature search and initial screening were conducted over a one-month period beginning on 15 October 2024.

A total of 1937 articles were initially retrieved from the database searches. During the first screening stage, titles and abstracts were reviewed to eliminate duplicates, non-Korean-language publications (e.g., in English or Japanese), literature reviews, qualitative studies, and research trend analyses. This process excluded 678 articles, leaving 1259 for further review. In the second screening stage, the introductions and RQs of the remaining 1259 articles were examined. Studies were excluded if they (a) did not target engineering students; (b) included engineering students but did not report results specific to them; (c) focused on instructional design, educational models, program implementation and evaluation (e.g., effects of capstone design or flipped learning), or the development of instruments for measuring affective–psychological or social traits. Following this stage, 135 articles met the criteria for further review. In the third and final screening stage, inclusion criteria were further refined to focus specifically on affective–psychological and social characteristics. Studies were excluded if they focused exclusively on career or employment-related variables, cognitive traits (e.g., creativity, learning styles, creative problem-solving), or unrelated variables such as dropout rates or course satisfaction. This resulted in the exclusion of 40 additional articles. Ultimately, 95 studies were selected for final analysis. The full inclusion and exclusion criteria and the study selection process are summarized in Table 1 and Table 2.

### 3.3. Literature Analysis Criteria

Trend analysis of a specific research domain typically involves examining factors such as publication year, research methodology, academic discipline, participant characteristics, research topics, and related variables. This study adopted three key criteria and employed content analysis to identify the major research themes within the selected literature.

First, a chronological analysis was conducted to examine temporal trends and developmental patterns in research on affective–psychological and social characteristics. Second, a journal analysis was performed to identify the disciplinary backgrounds of researchers contributing to this area of inquiry. Third, the selected studies were analyzed based on keywords, central themes, and recurring terminology related to learner characteristics. This stage involved the application of content analysis, a qualitative method for systematically analyzing textual data by identifying frequently occurring words, concepts, or themes. Content analysis enables data reduction and the extraction of core meanings, facilitating an organized synthesis of diverse studies ([118]).

Based on the findings, studies were classified into two primary domains: affective–psychological and social. Subcategories within each domain were then developed with reference to the psychological frameworks proposed by [109] ([109]) and [128] ([128]), to offer an integrated understanding of the learner characteristics of engineering undergraduates.

### 3.4. Literature Analysis Procedure

The collected studies were coded based on the following predetermined criteria: publication year, journal, research topic, and relevant variables. Thematic analysis was then conducted using research titles, abstracts, keywords, and full texts. If a single study addressed both affective–psychological and social characteristics, it was classified as an “integrated trait” study for analytical clarity. Ultimately, all studies were grouped into one of three categories: affective–psychological, social, or integrated trait studies.

Frequencies and percentages were calculated to summarize and visualize trends across the selected literature. To ensure the validity and reliability of the content analysis and classification process, the researcher conducted all stages manually and independently, without the use of automated software tools. In cases of ambiguity or uncertainty regarding classification, consultation was sought from a professor of education with expertise in systematic literature reviews, as many methodological principles—such as search strategies, inclusion and exclusion criteria, and transparent reporting—overlap between systematic and scoping reviews. This expertise was therefore relevant for enhancing the methodological rigor of this study. This collaborative review contributed to the consistency and credibility of the analysis.

Multiple strategies were implemented to enhance the rigor of the search and analysis procedures. These included the use of clearly defined inclusion and exclusion criteria, transparent documentation of the screening process, and iterative coding protocols. Additionally, an audit trail was maintained throughout the review process. The researcher also engaged in regular consultation with two experts in educational research methodology to reduce potential bias and improve interpretive accuracy.

## 4. Results

To address the RQs, the 95 selected studies were classified into three categories: affective–psychological, social, and integrated trait studies. The analysis proceeded in two stages. First, the general characteristics of the studies—including publication year and journal—were examined. Second, an in-depth thematic analysis was conducted to identify core research themes and related variables within each category.

### 4.1. Analysis of General Features of Research on Engineering Students’ Characteristics

The temporal distribution of the selected studies is illustrated in Figure 1. The earliest relevant publication appeared in 2008, and the number of studies increased steadily over time, peaking at 15 publications in 2019. A relatively high level of research activity was also observed in 2009 and 2010, with 8 publications each, and in 2022, which recorded 10 studies. These patterns suggest a sustained academic interest in the topic across the years. In contrast, research activity was comparatively limited in some years. Only one study was published in 2008, and just two studies appeared in 2023, indicating periods of relatively lower scholarly attention.

The distribution of the 95 selected studies across academic journals is summarized in Table 3. These articles were published in a total of 37 different journals. The analysis identified several journals with relatively high publication frequencies, indicating concentrated scholarly interest in specific outlets within the field.

A total of 95 studies on the characteristics of engineering students were published across 37 academic journals. Among these, the *Journal of Engineering Education Research* accounted for the largest share, with 41 articles (43.2%). This was followed by the *Journal of Korea Academia-Industrial Cooperation Society* and *The Journal of Learner-Centered Curriculum and Instruction*, each contributing four articles, and the *Journal of Practical Engineering Education* and *Asian Journal of Education*, each with three articles. Aside from the *Journal of Engineering Education Research*, no journal consistently published research in this area, indicating a relatively dispersed publication landscape.

### 4.2. Thematic Analysis of Research Findings on Engineering Students’ Characteristics

Based on the classification of learner characteristics, 38 studies (40%) focused on affective–psychological traits, 43 studies (45.3%) addressed social traits, and 14 studies (14.7%) examined integrated traits, encompassing both domains. These thematic categories were further analyzed to identify temporal trends in research focus, as illustrated in Figure 2.

#### 4.2.1. Affective–Psychological Characteristics

This section categorizes the thematic focus of studies addressing the affective–psychological characteristics of engineering students. A summary of the findings is presented in Table 4, along with the key variables examined, to provide an overview of prevailing research trends in this domain. The most frequently studied themes were self-efficacy and domain-specific efficacy, which appeared in 14 of the 38 studies (36.8%). These were followed by research on failure tolerance, stress, and coping strategies (eight studies, 21.1%). Other commonly explored topics included personality types (four studies, 10.5%), psychological well-being and psychological capital (four studies, 10.5%), and adaptation to university life (three studies, 7.9%). Less frequently addressed themes included mental health and depression (two studies, 5.3%), as well as cognitive style, grit, and questioning attitude, each examined in a single study (2.6%).

(1)Self-efficacy and Domain-specific Self-efficacy

Engineering students’ self-efficacy has been shown to vary by gender, timing of major selection, and creativity, and it positively influences academic and career-related outcomes ([19]). While male students tend to report higher general and engineering self-efficacy, no significant gender differences have been observed in actual academic performance ([60], [61]). High self-efficacy enhances task confidence, learning motivation, and problem-solving skills, consistent with [11]’s ([11]) theory that self-efficacy shapes individuals’ choices, effort, and persistence ([20]; [132]).

However, female students often perceive greater career barriers and report lower levels of self-efficacy and satisfaction, underscoring the need for gender-sensitive and sustained support strategies ([52]). Engineering-specific self-efficacy has been found to mediate the relationship between perceived career barriers and persistence ([92]) and to influence both persistence and career preparation ([90]). Other domain-specific forms of self-efficacy have also been explored. Engineering students generally report high levels of creative self-efficacy, though arts students occasionally report higher levels ([21]; [64]). Entrepreneurial self-efficacy significantly predicts entrepreneurial intentions ([134]), while writing self-efficacy has been shown to reduce writing anxiety and improve metacognitive strategy use ([45], [47]). These findings suggest that writing instruction for engineering students should incorporate emotional and strategic support to enhance writing confidence.

To foster sustained academic engagement and career readiness, it is essential to strengthen both general and domain-specific self-efficacy. Tailored educational interventions that promote competence, self-regulation, and strategic learning may support students’ holistic development. Future research should examine the causal relationships among various types of self-efficacy and explore their practical applications in engineering education contexts.

(2)Major Adjustment, Academic Adjustment, and Social Adaptation

Engineering students’ adjustment to university life is strongly influenced by the degree of alignment between their academic interests and chosen major. Misalignment has been shown to decrease motivation, satisfaction, and adaptability, while increasing the risk of withdrawal ([58]). Confidence in one’s academic choices also positively predicts adjustment outcomes ([78]), highlighting the need for more precise admissions processes and evaluative systems that assess students’ motivation and certainty at the time of enrollment. Among female engineering students, the lack of visible role models has emerged as a significant barrier to academic adjustment and engagement, contributing to increased perceptions of career-related obstacles ([25]). To address this, mentoring programs, exposure to successful role models, and structured exchanges with industry professionals have been recommended as effective support mechanisms to enhance adjustment and persistence.

(3)Sensing–thinking (ST) and Intuitive–thinking (NT) Personality Types

Engineering students in South Korea predominantly exhibit either ST or NT personality types ([15]; [93]). ST types are practical, detail-oriented, and prefer concrete, logical information, excelling in structured problem-solving tasks. In contrast, NT types are analytical and creative, characterized by a preference for abstract thinking and idea generation ([15]; [93]). These personality profiles underscore the importance of instructional approaches tailored to accommodate different cognitive styles. Additionally, judging and thinking traits have been found to significantly predict persistence in engineering majors ([93]), suggesting that both personality traits and academic aptitudes should be considered in curriculum design and student support strategies. Moreover, personality diversity within teams plays a critical role in collaborative performance. Research indicates that heterogeneous teams composed of students with varied personality types demonstrate higher creativity and more effective problem-solving than homogeneous groups ([3], [4]). These findings emphasize the value of personality-aware team formation in project-based learning environments, where diversity can be leveraged to enhance innovation and learning outcomes.

(4)External Thinking Style and Interpersonal Orientation

Compared to students in the humanities, social sciences, and the arts, engineering students tend to demonstrate a stronger external thinking style and a greater preference for interpersonal, collaboration-oriented problem-solving ([142]). Gender differences have also been observed: male engineering students are more likely to exhibit judicial and monarchic thinking styles, which emphasize analytical evaluation and single-goal focus, while female students are more inclined toward hierarchical thinking styles, characterized by the ability to prioritize and manage multiple goals ([142]). These findings suggest that thinking styles, influenced in part by gender, may shape how students engage in problem-solving, decision-making, and collaborative learning in engineering contexts. Understanding these differences has implications for instructional design, particularly in promoting inclusive classroom participation, diverse team formation, and adaptive learning strategies.

(5)Psychological Well-being and Psychological Capital

Psychological well-being and psychological capital are critical factors in engineering students’ academic adjustment and career development. Male students report higher levels of well-being across multiple dimensions, underscoring the need for gender-sensitive support programs ([40]). Psychological capital—comprising hope, resilience, and optimism—has been shown to significantly influence both academic satisfaction and career decision-making. In particular, hope, resilience, and satisfaction with one’s major are strong predictors of career indecision ([131]). These findings suggest that interventions grounded in positive psychology should be integrated into academic advising and career counseling to enhance students’ psychological resources, promote career maturity, and improve academic persistence. Moreover, students’ perceived academic competence and professors’ gender role expectations affect both major satisfaction and career aspirations. Interestingly, the presence of role models was found to influence the career outcomes of male students more significantly than those of female students ([24]), pointing to structural inequities in how gender roles shape career development. The development of psychological capital appears especially important for female students, whose career trajectories are shaped by a complex interplay of self-efficacy, career identity, and societal expectations. Targeted strategies that build psychological strengths such as hope and resilience may help mitigate these constraints.

In addition, psychological well-being is closely linked to writing-related variables. Among first-year engineering students, well-being correlates with metacognitive awareness and writing anxiety. Gender-sensitive instructional strategies in writing education—such as reducing anxiety in male students and enhancing metacognitive strategy use among female students—are therefore recommended to support student success ([39]).

(6)Grit and Academic Goal Achievement

Grit has been found to significantly impact academic performance and goal attainment among engineering students. Research indicates that differences in diligence and perfectionism correlate with varying levels of grit and academic aspiration ([95]). To support students across the grit spectrum, educational interventions such as goal-setting, self-reflective practices, and structured mentoring programs are recommended.

(7)Mental Health and Depression

Mental health challenges, including anxiety and depression, are critical concerns among engineering students. Anxiety has been shown to increase suicidal ideation and disrupt sleep quality, with excessive weekend sleep further exacerbating these issues ([77]). These findings underscore the urgent need for targeted psychological intervention programs that support mental well-being within engineering education. Mental health is closely linked to academic stress, highly competitive learning environments, and social isolation, all of which are prevalent in the South Korean engineering education context. Therefore, early identification of at-risk students and the establishment of emotionally supportive and integrated mental health systems are essential components of institutional support. Furthermore, autonomy and relationality have been found to mediate the relationship between depression and happiness ([68]). This suggests that programs aimed at promoting self-development and interpersonal skills may help alleviate depressive symptoms and enhance students’ overall well-being.

(8)Failure Tolerance, Stress, and Coping Strategies

Failure tolerance—defined as the tendency to respond constructively to failure ([22])—varies notably among engineering undergraduates, graduate students, and professionals. Undergraduate students, in particular, tend to demonstrate low levels of failure tolerance and are more likely to interpret failure as a negative outcome ([117]). Research indicates that goal orientation plays a critical role in shaping failure tolerance. Students with self-growth or prosocial contribution goals (both intrinsic and extrinsic) exhibit higher resilience in the face of failure. These findings suggest that fostering an educational climate that emphasizes social value awareness and intrinsic motivation can help engineering students simultaneously build professional competence and a sense of social responsibility. Failure tolerance also influences emotional regulation and resilience, both of which are closely linked to stress levels. In failure contexts, heightened vulnerability can increase anxiety and confusion, particularly during the career decision-making process. Engineering students often experience elevated career-related stress due to limited self-understanding, insufficient information, and external pressures ([85]), which distinguish them from peers in other academic disciplines. To address the psychological and career-related challenges faced by engineering students, career education that aligns students’ interests, aptitudes, vocational values, personality traits, and competencies has proven effective in facilitating career exploration and informed decision-making. Additionally, stress reduction within academic and research settings can be supported by promoting human rights awareness and fostering a human rights–friendly learning climate ([86]). Such environments enhance psychological safety, prevent traumatic academic experiences, and contribute to academic persistence. Establishing learning environments characterized by horizontal communication, mutual respect, and emotional support is therefore essential. Moreover, educational and counseling programs that enhance sensitivity to human rights are critical for student well-being. Engineering students report higher levels of employment-related stress than their peers in other majors ([99]), yet they exhibit limited use of effective coping strategies for career preparation and decision-making ([57]; [130]). These findings underscore the need for career and employment education tailored to the specific needs of engineering students, including the integration of career guidance into major-specific curricula. Furthermore, satisfaction with one’s academic major has been shown to mediate the relationship between job-seeking stress and depression ([116]), highlighting the importance of curricular and programmatic initiatives that enhance students’ connection to and satisfaction with their field of study. 

Although somewhat dated, a comparative study of engineering and medical students nearing graduation found that employment-related concerns were the primary source of stress for engineering students, whereas academic performance was the predominant stressor for medical students ([29]). In terms of coping strategies, engineering students more frequently relied on entertainment-based coping mechanisms, while medical students reported greater use of alcohol and tobacco. While these findings offer valuable insight into discipline-specific stress profiles, they must be interpreted with caution due to the age of the data ([29]; [99]). Significant shifts in the labor market, higher education systems, and social norms in recent years may have altered both the sources of stress and the coping strategies employed by students across disciplines.

(9)Emotion-based Questioning Attitudes

Engineering students’ questioning attitudes—ranging from active to passive or absent—are influenced by emotional factors such as confidence, anxiety, and avoidance tendencies ([56]), which are closely linked to the process of question generation. Passive or avoidant questioning behaviors are particularly associated with psychological anxiety and low self-efficacy. Therefore, emotionally supportive instructional strategies that encourage questioning can serve as effective interventions to foster more active engagement in inquiry-based learning.

#### 4.2.2. Social Characteristics

Social characteristics serve as the attitudinal and dispositional foundations upon which social competencies are cultivated and enacted. While these characteristics reflect internal orientations—such as responsibility, openness to others, and ethical awareness—they are expressed through externally observable skills, including communication, empathy, collaboration, and leadership. This section presents the thematic categories derived from content analysis of the 43 studies that focused on the social characteristics of engineering students. Key variables examined in these studies are summarized in Table 5 to illustrate broader research trends. Among the selected studies, core competencies, digital competencies, and convergence competencies emerged as the most frequently investigated themes, appearing in 14 of the 43 studies (32.6%). Other recurring topics included ethical awareness and values, addressed in seven studies (16.3%), and leadership and self-leadership, examined in six studies (14%). Empathy, communication competency, and teamwork competency were each explored in four studies (9.3%), while global mindset and multicultural competencies were discussed in two studies (4.7%).

(1)Low Empathy and Interpersonal Competence

One study found that female engineering students exhibit higher levels of empathy than their male counterparts, although both groups score lower than students in caregiving-related disciplines ([44]; [108]). Despite demonstrating strong emotional clarity and perspective-taking skills, engineering students tend to show relatively low affective empathy, indicating a need for targeted training in interpersonal competencies ([16]). Empathy has been positively associated with interpersonal competence, and humor has been identified as a mediating factor between internalized shame and interpersonal skills. These findings suggest that communication training programs incorporating humor may be effective in enhancing interpersonal competence among engineering students ([71]). 

(2)Communication Competency and Developmental Factors

Among engineering students, communication apprehension—defined as the anxiety, fear, or discomfort experienced in communicative contexts—is influenced by individual learning styles, with sensing and reflective learners reporting higher levels of anxiety ([69]). Creativity and thinking styles have also been found to significantly affect communication competence ([48], [49]). Furthermore, studies have identified associations between leadership, communication apprehension, and verbal aggression, highlighting the need for instructional strategies that integrate communication and leadership development while accounting for students’ cognitive and creative profiles ([36]).

(3)Teamwork Competency and Related Factors

Engineering students’ teamwork competency is significantly influenced by their thinking styles and creativity. Specifically, legislative, judicial, hierarchical, and global thinking styles have been shown to predict teamwork performance ([37]). Everyday creativity alone explained 35% of the variance in teamwork skills ([50]). Moreover, teamwork has been found to mediate the relationship between creative problem-solving and self-directed learning ([8], [9]), underscoring its central role in engineering education. These findings suggest that teamwork is not an isolated skill but a collaborative competency closely tied to higher-order thinking, creativity, and self-regulation. Therefore, engineering education should position teamwork as a core learning objective that integrates critical thinking, creative problem-solving, and learner autonomy. 

(4)Leadership and Self-leadership

Among engineering students, leadership encompasses creativity, openness to diversity, communication, active listening, trust-building, collaboration, goal-setting, and management skills ([53]). Team leadership has been shown to influence team commitment and performance through communication and trust, underscoring the importance of cultivating leadership within team-based learning environments ([70]). In contrast, self-leadership refers to the ability to autonomously regulate one’s thoughts, emotions, and behaviors in pursuit of goals. This trait has a significant impact on team project performance ([2]) and is positively associated with Type C cognitive dominance ([81]), creative problem-solving, social support, and interpersonal competence ([1]; [43]).

These findings suggest that leadership competency in engineering students extends beyond task execution to encompass the integrated development of psychological, cognitive, and social capabilities. Consequently, instructional design in engineering education should align individual reflection with collaborative task execution to promote both self-leadership and team leadership. Rotating leadership roles within project teams and integrating real-world, problem-based tasks through team-based learning (TBL) are recommended strategies to provide students with authentic leadership experiences.

(5)Core, Digital, and Convergence Competencies

To succeed academically and professionally, engineering students must develop a broad range of core competencies, digital literacy, and integrative thinking abilities. While they typically exhibit strong responsibility, task performance, and information-processing skills, they often show low levels of communication proficiency, intercultural understanding, and convergence competencies ([38]; [59]; [66]; [114]; [141]). Core competencies are significant predictors of academic achievement, with resource and information utilization and higher-order thinking skills strongly influencing performance in math, science, and computer courses ([38]; [34]). However, human resource professionals often prioritize soft skills and character over technical proficiency when evaluating engineering graduates ([72]). Notably, while graduates tend to view themselves as possessing strong job-related competencies, undergraduates assess their abilities primarily in relation to academic curricula, indicating a misalignment between educational outcomes and industry expectations that must be addressed. Digital competence, defined as the critical, effective, and ethical use of digital tools, includes skills in information retrieval, analysis, content creation, communication, and cybersecurity. It is increasingly emphasized at all educational levels and has been found to positively influence engineering students’ ability to identify entrepreneurial opportunities, with social network engagement moderating this relationship ([63]).

Research on integrative competencies among engineering students has produced mixed findings. Some studies report that engineering students perceive themselves as having stronger creative and convergent abilities compared to students in other majors ([62]; [65]), while others suggest that they exhibit relatively lower convergence competencies and require targeted instruction to foster creativity and interdisciplinary thinking ([14]). These competencies also vary by gender and academic year. For example, male students tend to score higher in areas such as knowledge creation and application, systems thinking, communication and collaboration, and future orientation ([33]; [79]). Furthermore, upper-year students generally demonstrate stronger convergence competencies, highlighting the effectiveness of progressive, scaffolded instructional models ([79]). Face-to-face instruction has been shown to be more effective than online learning in developing convergence capabilities, particularly due to the role of interpersonal interaction and collaboration ([79]). Additionally, factors such as major satisfaction and participation in accreditation programs have been positively associated with the development of convergence competencies ([89]).

In summary, engineering education must extend beyond the transmission of technical knowledge to emphasize integrated, competency-based instruction aimed at solving real-world and societal challenges. Curricular models should project-based learning, TBL, capstone design, and interdisciplinary, problem-centered courses to effectively cultivate both core and digital/convergence competencies.

(6)Global and Multicultural Competencies

Engineering students tend to demonstrate lower global adaptability but relatively higher levels of practical multicultural acceptance. Self-awareness has been identified as a key factor in fostering multicultural openness ([104]; [105]), suggesting that global and multicultural competencies extend beyond foreign language proficiency or cultural knowledge and can be cultivated through self-reflection and authentic intercultural engagement. Experiential learning opportunities—such as global issue-based projects, joint courses with international institutions, and multicultural team activities—can systematically enhance students’ international communication skills and cultural sensitivity. Furthermore, multicultural education, global leadership workshops, and orientation programs for student exchanges can contribute to strengthening global adaptability and promoting inclusive attitudes within engineering education.

(7)Ethical Awareness, Social Responsibility, and Values

While engineering students generally exhibit high awareness of academic ethics, ethical behavior in practice remains limited ([96]). To strengthen ethical awareness in engineering education, curricula should integrate information and communication ethics, ideally aligned with accreditation standards and frameworks ([97], [98]). Case-based instruction has also been recommended to enhance ethical sensitivity, particularly in emerging areas such as AI ethics ([87]).

In terms of social responsibility, engineering students show strong awareness of human welfare and environmental sustainability. However, their participation in public communication and civic engagement is relatively low, indicating a need for tailored educational approaches that encourage active social involvement ([83]). Regarding value formation, a significant correlation has been found between students’ sense of technological responsibility and their disciplinary interest, emphasizing the importance of discipline-specific ethics education grounded in real-world cases ([88]). Additionally, engineering students report higher awareness of physical, social, and mental health compared to peers in other fields, suggesting the value of health promotion programs designed to reflect both disciplinary and gender-specific needs ([13]).

(8)Social Capital and Collaborative Networks

Social capital refers to the resources individuals acquire through networks of trust, interaction, and social relationships within a broader social context. Among engineering students, social capital significantly influences knowledge-sharing intentions, defined as the intrinsic motivation and willingness to voluntarily exchange knowledge and information with others ([54]). To foster knowledge-sharing, educational environments should prioritize peer interaction, trust development, and the establishment of shared objectives. Instructional strategies such as team-based projects and structured mentoring programs can effectively enhance social capital and promote collaborative learning.

(9)Acceptance of Discipline-based Writing Instruction

Engineering students’ perceptions of the necessity of writing instruction vary by academic year and disciplinary context ([94]), indicating that writing education tailored to students’ disciplinary backgrounds and academic progression may enhance both receptivity and engagement. Contextualized writing instruction has been shown to increase acceptance and promote more meaningful participation in learning activities among engineering students.

#### 4.2.3. Integrated Characteristics

Studies examining both the affective–psychological and social characteristics of engineering students were thematically analyzed, as summarized in Table 6. Key variables were identified to provide a comprehensive overview of the research landscape. The most frequently explored topic was self-efficacy and achievement across academic, career, and occupational domains, addressed in 8 of the 14 studies (57.1%). This was followed by studies focusing on intrapersonal traits, interpersonal–creative competencies, and perceptions and attitudes toward social values, each represented in three studies (21.4%).

(1)Self-efficacy and Achievement in Academic, Career, and Occupational Domains

Engineering students’ self-efficacy, shaped by emotional and social support, is a key determinant of academic and career success. For instance, higher self-efficacy reduces speech anxiety and improves communication competence ([35]), indicating that communication training should not only address technical expression skills but also incorporate strategies to bolster self-efficacy and provide psychological support.

Among first-year students, self-efficacy, interest in engineering, and social support significantly influence adjustment and persistence ([91]). These findings suggest that fostering academic persistence requires enhancing self-efficacy, sustaining disciplinary interest, and supporting college adjustment through targeted emotional and institutional interventions.

Female engineering students’ career aspirations are shaped by self-efficacy and academic emotions, with role models playing a critical mediating role ([80]). The presence of female faculty influences how external factors enhance self-efficacy. In institutions with fewer female faculty, emotion-focused interventions are more effective, whereas in contexts with greater female representation, mentoring and role model-based strategies yield better outcomes. These findings underscore the importance of fostering gender-equitable educational environments and increasing the visibility of female role models to support the career development of female engineering students.

Contextual support also enhances students’ sense of belonging and disciplinary interest ([75]; [73]), with female students demonstrating greater responsiveness to support than to barriers. Therefore, promoting major retention and academic engagement is more effectively achieved by strengthening outcome expectations and providing emotional and social support, rather than solely targeting self-efficacy. Systematic interventions—such as mentoring, interpersonal engagement, and financial assistance—are particularly impactful for female students and should be implemented at both instructional and institutional levels.

Contextual support influences career indecision among engineering students through the mediating effects of coping self-efficacy and outcome expectations, with gender differences observed in the strength of these mediators. Specifically, male students are more affected by coping self-efficacy, whereas female students are more influenced by outcome expectations ([74]). These findings indicate the need for gender-sensitive career support strategies.

Students with higher levels of career decision-making exhibit stronger self-leadership and higher self-efficacy ([103]). Moreover, job search self-efficacy is negatively associated with perceived employment barriers and positively correlated with self-leadership ([133]). These results suggest that enhancing self-leadership and addressing structural employment barriers are critical for improving job search self-efficacy. Accordingly, strengthening engineering students’ career competencies requires targeted decision-making training, educational interventions to develop self-leadership, and institutional efforts to mitigate employment-related structural challenges.

(2)Intrapersonal Traits and Interpersonal–creative Competence

The personality traits and identity development of engineering students play a critical role in shaping their academic and career outcomes, as well as their interpersonal relationships and collaborative effectiveness.

Emotion regulation and interpersonal skills are central to stress adaptability and collaboration competence. Emotion regulation strategies—defined as the ability to modulate emotions in line with sociocultural norms under stress—significantly influence students’ capacity to manage interpersonal conflicts ([17]). To strengthen conflict resolution skills, training in core strategies such as sensory recovery, solution-focused thinking, and emotional clarity is essential. In this context, mindfulness-based stress reduction and emotion-focused therapy have been suggested as effective interventions.

Engineering students’ sense of agency is positively associated with self-awareness and identity development, while their sense of relationality correlates with group identity and interpersonal competence ([67]). Students with high levels of both agency and relationality tend to demonstrate stronger interpersonal skills, underscoring the importance of integrated character education that promotes personal maturity alongside collaboration-based competencies in engineering curricula.

Furthermore, engineering students report higher levels of creative convergence competence than peers in other disciplines. Among the “Big Five” personality traits, extraversion and conscientiousness significantly predict this competence ([115]), indicating the value of personality-informed educational strategies. To cultivate these abilities, engineering programs should foster creative learning environments and offer structured opportunities for meaningful collaboration.

(3)Perceptions and Attitudes toward Social Values

Perceptions of key employment factors and workplace success differ notably among engineering students, faculty members, and industry representatives ([113]). Employers tend to prioritize personality traits and affective competencies, such as attitudes and interpersonal skills, while students place greater emphasis on technical expertise, and faculty highlight problem-solving and proactiveness. Gender-based differences are also evident in students’ perceptions. These findings point to the need for gender-sensitive educational environments and systematic support structures within engineering education. For female students in particular, instructional strategies should strengthen responsibility, interpersonal competence, and adaptability to organizational culture. 

Engineering students’ attitudes toward sustainable development vary significantly by academic year, with fourth-year students demonstrating more mature perceptions of sustainability ([18]). This suggests that cumulative educational experiences and developmental maturity contribute to a deeper understanding of sustainability, supporting the need for curriculum design that aligns with students’ academic progression.

Regarding perceptions of the Fourth Industrial Revolution, female engineering students report greater concern about societal risks—such as AI misuse—whereas male students tend to express more optimism about its impact ([140]). Although underclassmen exhibit higher expectations for future technological changes, their level of actual preparedness does not significantly differ across academic years. These findings indicate that optimistic attitudes or expectations do not necessarily equate to readiness. Therefore, there is a clear need for customized education that accounts for both gender and academic level, as well as integrative curricula that address the ethical and societal implications of emerging technologies.

## 5. Discussion

This scoping review specifically examined studies on South Korean undergraduate engineering students, identifying characteristics unique to this group as well as patterns that may be generalizable to broader engineering education contexts. Unique traits observed among engineering students include the influence of engineering self-efficacy, the impact of career barriers on female students, the prevalence of ST and NT personality types, external thinking styles, and comparatively low empathy. In contrast, other characteristics—such as the relationship between major interest and academic adjustment—appear to reflect broader psychological, emotional, and social patterns common to university students in general. When situated within the academic and professional context of engineering education, these distinctive traits underscore the need for tailored educational interventions. At the same time, the more universal traits reinforce the importance of providing foundational psychological, emotional, and social support to engineering students throughout their academic and career development. Future research should seek to empirically distinguish engineering-specific traits from those shared across disciplines by incorporating appropriate comparison groups. Moreover, a more nuanced approach is needed to explore differences across engineering subfields.

The aim of this study was to systematically review and analyze research on the learner characteristics of South Korean engineering students and to identify key trends in the existing literature. A total of 95 studies were examined in terms of publication period, academic journal, research themes, and the variables addressed. The main findings and corresponding discussion are summarized below.

First, an analysis of the 95 selected studies by publication year revealed that the earliest study was published in 2008, with research activity gradually increasing and peaking in 2019 (15 studies). A total of 10 studies were published in 2022, while 2009 and 2010 each saw 8 publications, indicating sustained scholarly interest over time. These studies appeared across 37 academic journals, with the *Journal of Engineering Education Research* accounting for the largest share (41 articles, 43.2%). As prior studies analyzing research trends in engineering students’ learner characteristics are limited, direct comparisons and contextualization remain challenging. Nevertheless, the importance of accounting for learner characteristics—particularly in the design of effective teaching and learning strategies—has been widely emphasized in existing literature ([129]). In this context, the present study provides a systematic overview that may serve as a foundational resource for designing learner-centered instruction in engineering education. The findings also offer valuable insights for enhancing the educational experiences of engineering students and developing targeted support mechanisms.

Second, the studies were categorized according to the primary focus of learner characteristics: 38 studies (40%) addressed affective–psychological traits, 43 studies (45.3%) examined social traits, and 14 studies (14.7%) investigated both domains as integrated characteristics. In terms of thematic focus, studies on affective–psychological characteristics primarily examined general and domain-specific self-efficacy (14 studies, 36.8%), failure tolerance, stress, and coping strategies (8 studies, 21.1%), personality types (4 studies, 10.5%), and psychological well-being and psychological capital (4 studies, 10.5%). Other topics included adaptation (three studies, 7.9%), mental health and depression (two studies, 5.3%), and cognitive style, grit, and questioning attitudes (one study each, 2.6%). These findings suggest that both general and domain-specific self-efficacy are central motivational factors influencing learning and achievement among engineering students. However, the comparatively limited attention to the roles of affective and psychological traits in academic persistence and long-term career development indicates a need for further research. Given the specific demands of engineering education and professional pathways, students require support systems that simultaneously address academic motivation, career goal-setting, and psychosocial well-being. Structural factors such as the underrepresentation of female students, gender imbalances within engineering majors, and male-dominated industry cultures further underscore the need for targeted psychological and emotional interventions ([40], [41], [46]). These factors significantly affect academic persistence and career trajectories, particularly for female students. Accordingly, future research should more systematically examine how affective and psychological variables influence long-term academic engagement and career outcomes among engineering students.

The findings indicate that studies on social characteristics most frequently addressed core competencies, digital competencies, and convergence competencies (14 studies, 32.6%), followed by ethical awareness and values (7 studies, 16.3%), leadership and self-leadership (6 studies, 14%), and empathy, communication, and teamwork competencies (4 studies each, 9.3%). Global mindset and multicultural competence were addressed in only two studies (4.7%). These results suggest that practical competencies and collaborative skills relevant to professional performance are central themes in research on the social characteristics of engineering students. The increasing demand for global collaboration and interdisciplinary communication in the engineering profession further highlights the importance of these competencies. However, the review also reveals a notable gap in research on global and multicultural competencies, identifying a critical area for future investigation. Future studies should explore engineering students’ experiences in multicultural and cross-disciplinary teams, participation in global projects, and engagement in international problem-solving contexts. Correspondingly, educational strategies aimed at fostering global leadership and cultural adaptability warrant further development. Moreover, despite the growing societal expectations placed on engineers, studies focusing on ethical awareness and social responsibility (16.3%) remain limited. In the context of contemporary challenges—such as AI ethics, sustainability, and environmental impact—it is increasingly important for engineering education to cultivate ethical reasoning and a strong sense of social responsibility. Therefore, future research should investigate instructional approaches and evaluation frameworks that promote ethical decision-making and civic engagement among engineering students.

The results also indicate that most studies on integrated characteristics focused on the relationship between self-efficacy and academic, career, or employment outcomes (eight studies, 57.1%), followed by research on emotional, interpersonal, and personality traits, as well as awareness of social issues and value orientations (three studies each, 21.43%). These findings underscore the central role of self-efficacy in academic and career success but also highlight a relative lack of attention to how personality traits, identity formation, emotional regulation, and interpersonal competence influence educational and professional outcomes among engineering students.

In particular, there is a pressing need to examine how self-efficacy translates into measurable academic performance and job-related skills. Existing research insufficiently addresses the role of individual characteristics—such as personality and self-identity—within engineering education contexts. For instance, further investigation is needed into how personality traits relate to problem-solving, creative thinking, and teamwork, as well as how emotional regulation shapes students’ ability to manage academic stress and collaborate in project-based environments. Expanding the research agenda to include these dimensions will offer a more comprehensive understanding of the complex interplay between individual attributes and academic or professional achievement in engineering fields.

Third, the key variables and findings from studies on engineering students’ learner characteristics can be summarized as follows. 

First, among affective–psychological traits, self-efficacy emerged as a significant predictor of academic achievement, career decision-making, and problem-solving ability ([90]; [132]). Both creative and entrepreneurial self-efficacy were positively associated with creativity and career exploration ([64]; [134]), highlighting the need for educational interventions that strengthen self-efficacy to enhance academic and professional outcomes. South Korean undergraduate engineering students also commonly report high stress and mental health challenges due to heavy academic workloads and career uncertainty ([116]), underscoring the importance of psychological support mechanisms grounded in social relationship-building within the Korean higher education context ([68]). Personality traits significantly influence academic adjustment, team performance, and teamwork competencies ([93]), while personality diversity has been shown to improve team creativity and problem-solving ([3], [4]). In terms of thinking styles, engineering students tend to prefer external styles, with male students showing a stronger inclination toward judicial and monarchic styles, and female students favoring hierarchical styles ([142]). These findings point to the importance of implementing tailored instructional and team-based strategies that reflect students’ personality profiles and cognitive styles. Finally, psychological well-being is closely associated with academic and major satisfaction as well as persistence ([40], [41]). Along with metacognition, it also predicts writing anxiety ([39]), suggesting a need for emotional support programs and instructional approaches that promote both emotional stability and academic engagement.

Second, regarding social characteristics, engineering students generally demonstrate lower levels of empathy but higher systematization abilities compared to students in other disciplines, suggesting a strength in analytical and structured problem-solving ([108]). However, their limited emotional empathy ([16]) highlights the need for targeted interventions. Convergence-oriented attitudes have been found to support empathy development, indicating the potential of integrative education approaches in addressing this gap ([43]). Core competencies, which are essential for both academic and professional success, have been a key focus of research. Notably, communication and convergence competencies are often underdeveloped among engineering students ([38]; [59]; [66]; [114]; [141]). Digital competence, including the ability to critically and ethically engage with digital tools, is positively associated with entrepreneurial opportunities and career success ([63]). Similarly, communication skills are linked to learning attitudes, creativity, and cognitive styles ([48], [49]; [69]), reinforcing the need for enhanced training in both digital literacy and communication. Teamwork and leadership are also vital for collaborative and creative problem-solving ([9]; [50]; [70]). Therefore, sustained use of team- and role-based instructional strategies is recommended. Finally, ethical awareness and social responsibility—critical to both academic integrity and professional conduct—require ethics education grounded in case-based learning ([97], [98]; [96]; [87]).

Third, studies addressing mixed characteristics—encompassing both affective–psychological and social dimensions—have demonstrated that self-efficacy and motivation significantly influence academic performance, career development, interpersonal skills, and communication competence ([35]; [73]; [80]; [75]; [91]). For instance, job search self-efficacy is negatively associated with perceived career barriers and positively associated with self-leadership ([133]), highlighting the importance of programs that strengthen self-efficacy to enhance persistence and career preparedness. Moreover, personality traits have a substantial impact on both academic and professional achievement, as well as on creative convergence competence ([115]), underscoring the value of personality-informed instructional strategies in convergence education. Self-identity is also closely linked to interpersonal competence ([67]), indicating the need for integrated programs that promote both personal development and social cohesion. These findings suggest that engineering education should not only cultivate technical expertise but also support identity formation and interpersonal growth. Additionally, emotion regulation strategies contribute to conflict resolution in both academic and social settings ([17]), affirming the role of emotional competence in facilitating effective team interactions.

## 6. Implications for Future Research and Engineering Education

This study yields several key implications for future research and practice in engineering education. First, self-efficacy emerged as the most frequently examined variable in studies on affective–psychological and integrated learner characteristics. This finding aligns with social cognitive career theory ([100]), which emphasizes the dynamic interplay of self-efficacy, outcome expectations, contextual supports, academic persistence, and career exploration. A substantial body of research confirms that engineering students’ self-efficacy strongly influences academic achievement, persistence, and problem-solving ability, underscoring its importance as a central focus for both research and educational interventions. Additionally, social competencies—such as core competencies, teamwork, and convergence ability—have been identified as key predictors of learning engagement and professional success. Future research should therefore explore how motivation and self-efficacy interact with variables such as gender and academic level to shape learning outcomes and career development. However, this review also reveals notable research gaps. Characteristics such as resilience, academic burnout, emotional exhaustion, intrinsic motivation, metacognition, cognitive flexibility, and emotional regulation remain underexplored in the engineering education context. Similarly, social factors—including social responsibility, networking for collaborative work, leadership, and followership in team-based tasks—require further empirical investigation. These areas are particularly critical given the growing emphasis on the societal and ethical dimensions of engineering practice.

Second, although numerous studies have emphasized the importance of instructional design and programs tailored to learners’ psychological and social traits, few have presented evidence-based models that are implemented and evaluated in actual educational settings. Future research should therefore prioritize the empirical validation of such interventions to inform the development of sustainable and effective educational models in engineering.

Third, the majority of existing studies rely on cross-sectional data, offering only static snapshots of students’ development. To better understand the evolving nature of affective and social traits and their cumulative effects on learning outcomes and professional readiness, longitudinal research is essential. Furthermore, the limited sample sizes in some studies restrict the generalizability of their findings, underscoring the need for more robust research designs and larger, more diverse participant groups.

Fourth, while engineering students typically excel in analytical and logical reasoning, they often exhibit underdeveloped emotional empathy and limited interpersonal conflict resolution skills. This highlights the need for further research on interventions that enhance emotional regulation, teamwork, and communication abilities. Relevant programs may include scientific writing instruction, presentation training, and interpersonal skills development. Additionally, creative convergence competencies—strongly associated with personality traits such as extraversion and conscientiousness—should be cultivated through interdisciplinary projects, entrepreneurship initiatives, and integrated art–engineering curricula. Given that female engineering students report lower levels of self-efficacy, belonging, and career aspiration, it is critical to develop gender-sensitive models and offer sustained support mechanisms, such as mentoring programs and women-in-engineering networks.

Finally, in the context of rapid AI and digital transformation, the integration of digital literacy, data analysis, and AI education into the engineering curriculum should be prioritized. Future studies should also examine the impact of these competencies on learning outcomes and career preparedness.

## 7. Limitations and Future Discussion

This study was limited to academic journal articles published in South Korea, which may constrain the generalizability of the findings. To achieve a more comprehensive and cross-cultural understanding of engineering students’ characteristics, future research should incorporate dissertations and international literature. Additionally, many of the analyzed studies were based on single-institution samples, necessitating caution in extending the results to broader populations. There is also a possibility that relevant studies were omitted due to keyword or database limitations. To enhance reliability and representativeness, future reviews should employ more systematic and inclusive data collection strategies. Despite these limitations, the present study provides valuable insights into the affective–psychological and social characteristics of South Korean engineering students and offers a solid foundation for learner-centered educational planning and future research.

## 8. Conclusions

This scoping review provides a comprehensive synthesis of research on the affective–psychological and social characteristics of engineering students in South Korea. By analyzing 95 studies across diverse themes and variables, the review identified both distinctive and generalizable traits within this population. Key findings underscore the pivotal role of self-efficacy in academic persistence and career development, as well as the importance of social competencies such as teamwork and communication. Additionally, the review highlights the need to address underexplored areas, including emotional regulation, resilience, and ethical awareness. Notable gaps include the limited empirical validation of educational interventions, the predominance of cross-sectional study designs, and a lack of research addressing gender issues and digital transformation in engineering education.

Overall, the findings offer foundational insights for developing learner-centered, evidence-based strategies to support the academic success and holistic development of engineering students. Continued empirical research is essential to advance inclusive, future-oriented engineering education that fosters not only technical proficiency but also the emotional, social, and ethical competencies required in an increasingly complex and globalized world.

## Figures and Tables

**Figure 1 behavsci-15-01189-f001:**
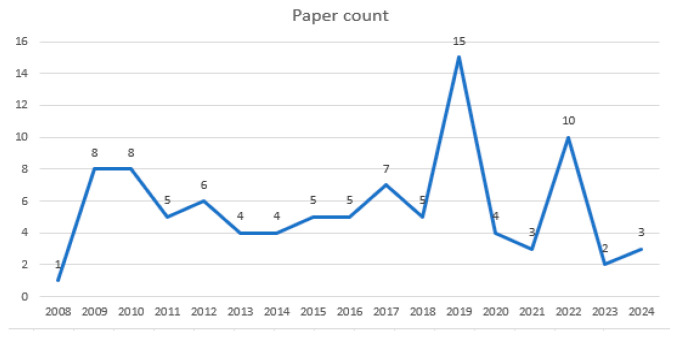
Targeted studies by publication year.

**Figure 2 behavsci-15-01189-f002:**
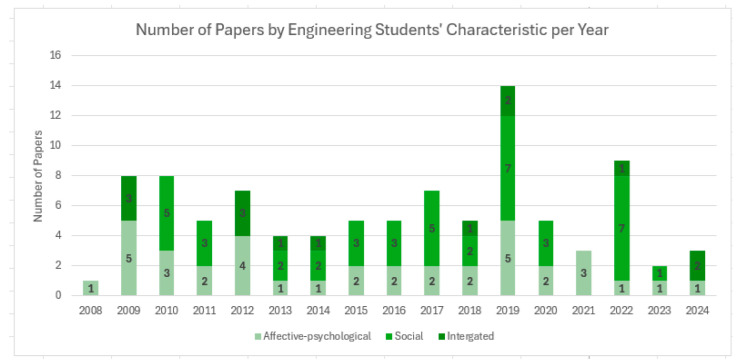
Distribution of publications by engineering students’ characteristics.

**Table 1 behavsci-15-01189-t001:** Eligibility Criteria.

Stage	Exclusion Criteria
Stage 1	Duplicate publications Articles written in foreign languages (e.g., English and Japanese) Literature reviews, qualitative studies, and trend analyses
Stage 2	Studies that did not include engineering students as part of the research sample Studies that included engineering students but did not report results specifically for them Studies focused on instructional design, models, curriculum development, or educational program design Studies analyzing educational effectiveness (e.g., effects of flipped learning) Studies focused on developing instruments to measure affective–psychological and social characteristics
Stage 3	Studies addressing only the career- or employment-related variables of engineering students Studies focused exclusively on the cognitive traits of engineering students Studies unrelated to affective–psychological and social characteristics

**Table 2 behavsci-15-01189-t002:** Screening process based on PRISMA guidelines.

Literature Identification		Search Results Using Keyword Combinations (*N* = 1937)		
Screening		Stage 1: Title and abstract review *(N = 1937)*	→	Excluded: 678 studies
		Stage 2:	→	Excluded: 1124 studies
		Review of introductions and RQs *(N = 1259)*		
		Stage 3: In-depth content review *(N = 135)*	→	Excluded: 40 studies
Literature analysis		Total studies included in final analysis *(N = 95)*		

**Table 3 behavsci-15-01189-t003:** Number of published papers by academic journals.

Journal Name (37 Journals)	Number of Articles Published by Journal
*Journal of Engineering Education Research*	41
*Journal of Korea Academia-Industrial cooperation Society*	4
*The Journal of Learner-Centered Curriculum and Instruction*	4
*Journal of Practical Engineering Education*	3
*Asian Journal of Education*	3
*Korean Journal of General Education*	2
*Korean Journal of Educational Research*	2
*The Journal of Humanities and Social science*	2
*The Journal of Vocational Education Research*	2
*Korean Journal of Youth Studies*	2
*Journal of Curriculum Integration*	2
*Journal of Knowledge Information Technology and Systems*	2
*Journal of the Korean Data Analysis Society*	2
Others	24
Total	95

**Table 4 behavsci-15-01189-t004:** Key variables and analysis of studies on affective–psychological characteristics (*N* = 38).

Research Topic	Independent Variables	Dependent Variables	Research
General and domain-specific self-efficacy	Self-efficacy	Motivation for major selection	[19] ([19])
	Engineering self-efficacy	Learning outcomes and achievement	[60] ([60])
	Self-efficacy, engineering self-efficacy	Academic achievement	[61] ([61])
	Self-efficacy	Problem-solving ability	[132] ([132])
	Learning motivation	Learning behavior and satisfaction	[20] ([20])
	Competitive motivation	Learning strategies	[76] ([76])
	Psychological characteristics related to career barriers (e.g., perceived value of the major, career-related expectations, aspirations for major-related career experience, satisfaction with the major, major-related self-efficacy, and self-efficacy for multiple roles)		[52] ([52])
	Career barriers	Academic persistence intention (mediator: engineering self-efficacy)	[92] ([92])
	Engineering self-efficacy	Academic persistence and career preparation behavior (mediators: outcome expectations, interest)	[90] ([90])
	Creative self-efficacy and creative problem-solving ability	–	[64] ([64])
	Creative self-efficacy	–	[21] ([21])
	Entrepreneurial self-efficacy and entrepreneurial orientation	Entrepreneurial intention	[134] ([134])
	Writing self-efficacy and metacognitive strategies	Writing anxiety	[45] ([45])
	Self-regulated learning strategies	Writing self-efficacy, writing feedback perception, sense of learning presence	[47] ([47])
Major adaptation, academic adjustment, and social adaptation	Interest, major fit, major adaptation	–	[58] ([58])
	Certainty in major selection, social adaptation, and academic adaptation	–	[78] ([78])
	Role model	Academic psychological variables (academic competence, course adaptation, course engagement)	[25] ([25])
Personality types	Personality type	–	[15] ([15])
	Personality type	Academic performance	[93] ([93])
	Personality (heterogeneous against homogeneous teams)	Team creativity and creative problem-solving ability	[3] ([3])
	Personality (heterogeneous against homogeneous teams)	Team creativity	[4] ([4])
Thinking styles	Thinking styles	–	[142] ([142])
Psychological well-being and psychological capital	Psychological well-being	Major satisfaction	[40] ([40])
	Psychological capital and academic satisfaction	Career indecision	[131] ([131])
	Internal psychological factors (academic competence, perceived autonomy, social skills, stress coping skills)	Psychological factors related to major courses (major satisfaction, course adaptation)	[24] ([24])
	Perceived environmental factors (gender role expectations of instructors and peers, role models, sense of belonging)	Career-related psychological factors (career aspiration, career-related self-efficacy)	
	Psychological well-being and metacognition	Writing anxiety, writing ability	[39] ([39])
Grit and academic goal attainment	Grit, life goals, conscientiousness	–	[95] ([95])
Mental health and depression	Anxiety and sleep quality	Suicidal ideation	[77] ([77])
	Depression	Happiness (mediators: sense of agency, relationships)	[68] ([68])
Failure tolerance, stress, and coping strategies	Social concern goals	Failure tolerance (persistence in the face of failure)	[117] ([117])
	Emotional and personality traits and difficulty in career decision-making	Emotional/personality-related career problems	[85] ([85])
	Perception of human rights situations	Stress (mediator: traumatic impact)	[86] ([86])
	Job-seeking stress and career decision-making confidence	–	[99] ([99])
	Stress coping strategies	Career preparation behavior	[57] ([57])
	Stress, depression	Adaptation to college life	[130] ([130])
	Job-seeking stress	Depression (mediator: major satisfaction)	[116] ([116])
	Depression and employment-related issues	–	[29] ([29])
Emotion-based questioning attitude	Questioning attitude	–	[56] ([56])

**Table 5 behavsci-15-01189-t005:** Key variables and analysis of studies on social characteristics (*N* = 43).

Research Topic	Independent Variables	Dependent Variables	Research
Empathy	Convergent attitude	Empathy	[44] ([44])
	Empathy, systematization ability	–	[108] ([108])
	Empathy	Interpersonal problems and emotional clarity	[16] ([16])
	Internalized shame	Interpersonal competence (mediators: empathy, sense of humor)	[71] ([71])
Communication competency and developmental factors	Learning styles	Communication apprehension	[69] ([69])
	Everyday creativity	Communication competence	[48] ([48])
	Thinking styles	Communication competence	[49] ([49])
	Leadership, communication apprehension, communication style	–	[36] ([36])
Teamwork competency and related factors	Thinking styles	Teamwork competency	[37] ([37])
	Everyday creativity	Teamwork competency	[50] ([50])
	Teamwork competency	Problem-solving ability (mediator: creative personality)	[8] ([8])
	Everyday creativity	Problem-solving ability (mediators: teamwork competency, self-directed learning ability)	[9] ([9])
Leadership and self-leadership	Leadership Performance Level	–	[53] ([53])
	Team leadership	Team effectiveness (team performance, team commitment) (mediator: team processes—communication, conflict, trust)	[70] ([70])
	Self-leadership	Types of creative problem-solving and locus of control	[1] ([1])
	Thinking styles and self-leadership	–	[81] ([81])
	Learning styles and self-leadership	Team performance grades	[2] ([2])
	Social support and interpersonal relationships	Self-leadership	[43] ([43])
Core competencies, digital competencies, and convergence competencies	Core life competencies and responsibility execution competency	–	[59] ([59])
	Core competencies and personal variables (gender, major, academic year, academic achievement)	–	[66] ([66])
	Core competencies, knowledge information-processing competency	–	[114] ([114])
	Core competencies	–	[141] ([141])
	Core competencies and extracurricular activities	–	[38] ([38])
	Core competencies	Academic performance (grade point average, GPA)	[34] ([34])
	Importance of core competencies	–	[72] ([72])
	Digital competency	Entrepreneurial opportunity competency (moderator: social network)	[63] ([63])
	Creative convergence competency	–	[62] ([62])
	Creative convergence competency	–	[65] ([65])
	Core competencies (communication, creativity, convergence)	–	[14] ([14])
	Major satisfaction	Creative convergence competency	[79] ([79])
	Engineering convergence competency	–	[33] ([33])
	Convergent attitude	–	[89] ([89])
Global and multicultural competencies	Global mindset, understanding of foreign cultures, and self-directed adaptability	–	[104] ([104])
	Multicultural acceptance, individual factors, family factors, and school factors	–	[105] ([105])
Ethical awareness, social responsibility, and values	Learning ethics	–	[96] ([96])
	Ethical awareness	–	[98] ([98])
	Information and communication ethics	–	[97] ([97])
	Ethical sensitivity	–	[87] ([87])
	Scientific and technological ethics and values	–	[88] ([88])
	Social responsibility	–	[83] ([83])
	Wellness and health awareness	–	[13] ([13])
Social capital and collaborative networks	Perception of social capital (structural, relational, and cognitive dimensions)	Intention to share knowledge	[54] ([54])
Acceptance of discipline-based writing instruction	Perception of writing courses	–	[94] ([94])

**Table 6 behavsci-15-01189-t006:** Key variables and analysis of studies on integrated characteristics (*N* = 14).

Research Topic	Independent Variables	Dependent Variables	Research
Self-efficacy and (academic, career, and job) achievement	Self-efficacy	Speech anxiety and communication competence	[35] ([35])
	Self-efficacy, interest in major, social support	College adjustment and academic persistence	[91] ([91])
	Influence of others	Career aspirations (mediators: positive academic emotions, engineering self-efficacy)	[80] ([80])
	Contextual support	Interest in engineering (mediators: academic self-efficacy, outcome expectations)	[75] ([75])
	Contextual support	Sense of major belonging (mediators: engineering self-efficacy, outcome expectations)	[73] ([73])
	Contextual support	Career indecision (mediators: coping efficacy, outcome expectations)	[74] ([74])
	Self-leadership, level of career decision, career preparation behavior	Self-efficacy	[103] ([103])
	Self-leadership and perceived employment barriers	Job search self-efficacy	[133] ([133])
Intrapersonal characteristics and interpersonal–creative capacities	Emotion regulation strategies	Conflict management strategies in interpersonal relationships	[17] ([17])
	Sense of agency, relationality, identity awareness, interpersonal competence	–	[67] ([67])
	Big Five personality traits	Creative convergence competence	[115] ([115])
Perceptions and attitudes toward social values	Perception of success factors	–	[113] ([113])
	Attitudes toward sustainable development	–	[18] ([18])
	Perception of the Fourth Industrial Revolution	–	[140] ([140])

## Data Availability

Data are available from the corresponding author upon reasonable request.

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
