# Peer review of "Scoping Review of Studies on Affective–Psychological and Social Characteristics of South Korean Engineering Students"

_behavsci, 2025, doi:10.3390/bs15091189_

Round 1
Reviewer 1 Report
Comments and Suggestions for Authors
It seems that the author has been very meticulous in designing and implementing this study. Good job on documenting the methodology in this scoping review.
I just have a few areas of minor adjustments recommended:
1) First sentence - "Science, technology, engineering, and mathematics higher education is globally recognized as a strategic pathway for cultivating future industry leaders." Should be "Science, technology, engineering, and mathematics in higher education is globally recognized as a strategic pathway for cultivating future industry leaders."
2) I know what the author is trying to say by using the label "noncognitive characteristics", but technically speaking, social processes (and arguably some affective processes) involve some cognition. Perhaps a better term would be "processes outside of academic intellectual aptitude". The modified term doesn't have to be these exact words, but I think it should be a term that better distinguishes what the author is trying to say (since there are cognitive elements in social and some affective processes).
3) The last 2 paragraphs in "Discussion" start with "Second" and "Third" respectively. This is after the 3rd last paragraph begins with "Third". Is this an overlooked error? Should the last 2 paragraphs of this section begin with "Fourth" and "Fifth"?
Author Response
Response to Reviewer’s Comments
We sincerely appreciate the reviewers' valuable comments and have thoroughly revised the manuscript to address their feedback to the best of our ability. Responses to the reviewers' comments are provided, and the revised sections of the manuscript are highlighted in yellow.
Overall Comment
It seems that the author has been very meticulous in designing and implementing this study. Good job on documenting the methodology in this scoping review.
I just have a few areas of minor adjustments recommended:
Response: We sincerely thank the reviewer for the positive and encouraging feedback on the design and implementation of this study. We truly appreciate the acknowledgement of the detailed documentation of our methodology in this scoping review. We also value the reviewer’s constructive suggestions for minor adjustments. All the recommended changes have been carefully considered and incorporated into the revised manuscript.
- Comment 1
First sentence wording: “Science, technology, engineering, and mathematics higher education is globally recognized …” → should be “Science, technology, engineering, and mathematics in higher education is globally recognized …”
Response: We appreciate the reviewer’s careful observation. We have revised the sentence accordingly to improve grammatical accuracy.
Revision: “Science, technology, engineering, and mathematics in higher education is globally recognized as a strategic pathway for cultivating future industry leaders.”
- Comment 2
The label “noncognitive characteristics” may not be appropriate since social and affective processes involve cognition. A more precise term is recommended, such as “processes outside of academic intellectual aptitude.”
Response: We appreciate the reviewer’s insightful comment regarding the use of the term “noncognitive characteristics.” We agree that social and affective processes inherently involve cognition, and thus the label noncognitive may lead to conceptual ambiguity. To address this concern, we have revised the manuscript to consistently use the term “affective–psychological and social characteristics” instead. In doing so, we aim to provide a more precise and theoretically appropriate description of the constructs under investigation.
Revision: p. 2. The revised content is presented as follows.
Against this backdrop, the present scoping review aims to examine academic literature on the affective–psychological and social characteristics of undergraduate engineering students in South Korea.
In engineering education, recent scoping reviews have addressed a range of affective–psychological and social domains and instructional approaches aligned with the present study’s focus.
- Comment 3
The last two paragraphs in “Discussion” start with “Second” and “Third” respectively, even though an earlier paragraph also begins with “Third.” Should these be “Fourth” and “Fifth” instead?
Response: We sincerely thank the reviewer for carefully noticing this inconsistency. We have revised the numbering in the “Discussion” section to ensure logical consistency. Specifically, we clarified the sequence of the three domains of engineering students’ learner characteristics—First (affective–psychological characteristics), Second (social characteristics), and Third (mixed characteristics). This correction eliminates the earlier duplication of “Third” and provides a coherent structure throughout the section.
Revision: p. 22
The paragraphs in the “Discussion” now begin with:
- First, among affective–psychological characteristics …
- Second, regarding social characteristics …
- Third, studies addressing mixed characteristics …
We would like to sincerely thank the reviewers once again for their careful reading, insightful comments, and constructive suggestions. Their feedback has been invaluable in improving the clarity, precision, and overall quality of our manuscript. We believe that the revisions made in response to the reviewers’ comments have strengthened the contribution of this study.

Reviewer 2 Report
Comments and Suggestions for Authors
The manuscript is well organized, and the purpose is of utmost importance given, as you indicated, the rising demands of artificial intelligence. Thus, exploring the literature on engineering students affective–psychological and social characteristics, rather than cognitive outcomes, is valuable for researchers and faculty alike.
One thing that may strengthen your introduction is to identify which of these citations e.g., paragraphs 2 & 3 use South Korea as a study setting as you indicated the academic rigor of South Korean institutions. The same notion goes for your literature review section.
Within your study methodologies section, it is unclear if your analyses were conducted by hand or if you used a software. Additionally, you write "onsultation was sought from
a professor of education with expertise in systematic literature reviews." Why did you seek out someone with expertise in systemic literature reviews rather than scoping reviews? Explaining this will enhance this section.
Your results section is very thorough, and you present findings using a balanced combination of tables, graphics, and narrative. I also greatly appreciate the discussion and limitations.
Finally, your research reveals the importance of affective–psychological and social characteristics and the need to explore further.
Author Response
Response to Reviewer’s Comments
We sincerely appreciate the reviewers' valuable comments and have thoroughly revised the manuscript to address their feedback to the best of our ability. Responses to the reviewers' comments are provided, and the revised sections of the manuscript are highlighted in yellow.
Overall Comment
The manuscript is well organized, and the purpose is of utmost importance given, as you indicated, the rising demands of artificial intelligence. Thus, exploring the literature on engineering students affective–psychological and social characteristics, rather than cognitive outcomes, is valuable for researchers and faculty alike.
Response:
We sincerely thank the reviewer for the positive and encouraging feedback on the organization, clarity, and importance of our manuscript. We are also grateful for the reviewer’s thoughtful suggestions to improve the introduction, methodology, and literature review sections. We have carefully revised the manuscript in response to each of the comments, as outlined below.
- Comment 1
One thing that may strengthen your introduction is to identify which of these citations e.g., paragraphs 2 & 3 use South Korea as a study setting as you indicated the academic rigor of South Korean institutions. The same notion goes for your literature review section.
Response: We thank the reviewer for this valuable suggestion. We agree that specifying which studies were conducted in the South Korean context enhances the transparency and relevance of the review.
Revision: In both the Introduction (paragraphs 2 and 3) and the Literature Review section, we have now explicitly indicated which citations involve studies conducted in South Korea. This allows readers to better contextualize the discussion with respect to the academic rigor of South Korean higher education institutions.
- Comment 2
Within your study methodologies section, it is unclear if your analyses were conducted by hand or if you used a software.
Response: We thank the reviewer for pointing out the need for clarification. All coding, classification, and thematic analysis were conducted manually by the researcher, rather than using any software package. We have revised the methodology section to explicitly indicate this, ensuring greater transparency.
Revision: p. 7. In the Methodology section, we have now specified as follows:
To ensure the validity and reliability of the content analysis and classification process, the primary researcher conducted all stages manually and independently, without the use of automated software tools.
3. Comment 3
Additionally, you write "consultation was sought from a professor of education with expertise in systematic literature reviews." Why did you seek out someone with expertise in systemic literature reviews rather than scoping reviews? Explaining this will enhance this section.
Response: We thank the reviewer for this insightful question. We recognize the need to clarify the rationale for consulting an expert in systematic rather than scoping reviews.
Revision: p. 7. We have now explained in the Methodology section that the consultation was sought from a professor of education with expertise in systematic literature reviews because many methodological principles overlap between systematic and scoping reviews. In particular, issues of search strategy, inclusion/exclusion criteria, and transparency of reporting are common to both approaches. We have clarified this reasoning in the manuscript to avoid confusion.
4. Comment 4
Your results section is very thorough, and you present findings using a balanced combination of tables, graphics, and narrative. I also greatly appreciate the discussion and limitations.
Response: We are grateful for the reviewer’s positive assessment of the Results and Discussion sections.
- Comment 5
Finally, your research reveals the importance of affective–psychological and social characteristics and the need to explore further.
Response: We sincerely thank the reviewer for highlighting the contribution of this study. We have emphasized this point more explicitly in the Conclusion section, underscoring the importance of future research on affective–psychological and social characteristics of engineering students.
We would like to sincerely thank the reviewers once again for their careful reading, insightful comments, and constructive suggestions. Their feedback has been invaluable in improving the clarity, precision, and overall quality of our manuscript. We believe that the revisions made in response to the reviewers’ comments have strengthened the contribution of this study.

Reviewer 3 Report
Comments and Suggestions for Authors
Generally the manuscript is very well-written, clear and transparent. the mapping of the themes is very useful and appropriate and the very narrow and specific scope of the review (at least in terms of context) makes its application and usefulness very clear.
Major issues:
- I think the authors should report a PRISMA-ScR checklist to help the reader navigate your study. The authors should find specific PRISMA reporting guidelines that also adhere to the MDPI expectations of transparency and apply them.
- The study needs a reading through and alignment of who and how is impacts. The study is very specific in its focus on south korean undergraduate engineerings students, which makes it highly relevant and interesting for specifically that group pf students, but that is very important to specify whenever you conclude/discuss about the results. an example is in the abstract where the authors write: "This review offers foundational insights for designing inclusive, future-oriented educational programs tailored to the diverse needs of engineering students." which i find a bit far reaching given the specific scope of the review.
Minor issues:
- The abstract could use a sentence about included databases.
- On page 5 there is something wrong with the numbering. The numbering should start one row lower than it does currently.
- There are some minor terminology errors that should be fixed (e.g. "affective-psychological" vs "affective/psychological").
- Some of the tables' captions are missing explanations of categories and denominators.
Author Response
Response to Reviewer’s Comments
We sincerely appreciate the reviewers' valuable comments and have thoroughly revised the manuscript to address their feedback to the best of our ability. Responses to the reviewers' comments are provided, and the revised sections of the manuscript are highlighted in yellow.
Overall Comment
Generally the manuscript is very well-written, clear and transparent. the mapping of the themes is very useful and appropriate and the very narrow and specific scope of the review (at least in terms of context) makes its application and usefulness very clear.
Response: We sincerely thank the reviewer for the positive and encouraging evaluation of our manuscript. We greatly appreciate the acknowledgement that the manuscript is well-written, clear, and transparent, and that the mapping of themes and the specific scope of the review contribute to its usefulness. We also value the reviewer’s constructive feedback and have carefully revised the manuscript in line with all suggestions.
- Comment 1
The authors should report a PRISMA-ScR checklist to help the reader navigate your study. The authors should find specific PRISMA reporting guidelines that also adhere to the MDPI expectations of transparency and apply them.
Response: We thank the reviewer for this important suggestion. We agree that including the PRISMA-ScR checklist will strengthen transparency and adherence to MDPI’s expectations.
Revision: We have added the completed PRISMA-ScR checklist as supplementary material and explicitly referenced it in the Methods section to help readers navigate our study more transparently. We have specified as follows:
“In addition, this review adhered to the PRISMA-ScR extension for scoping reviews (Tricco et al., 2018), in line with MDPI’s expectations of transparency and reporting standards. The completed PRISMA-ScR checklist is provided in Supplementary File 1.”
2. Comment 2
The study needs a reading through and alignment of who and how is impacts. The study is very specific in its focus on south Korean undergraduate engineering students, which makes it highly relevant and interesting for specifically that group pf students, but that is very important to specify whenever you conclude/discuss about the results. an example is in the abstract where the authors write: “This review offers foundational insights for designing inclusive, future-oriented educational programs tailored to the diverse needs of engineering students.” which I find a bit far reaching given the specific scope of the review.
Response: We thank the reviewer for this important observation. We agree that the conclusions and implications of this review should more explicitly reflect its specific focus on South Korean undergraduate engineering students. To address this, we have revised the abstract, discussion, and conclusion sections to consistently emphasize the South Korean context and to avoid overgeneralization to engineering students in general.
Revision: In the Abstract and Discussion, we have revised the language to emphasize that the findings are most applicable to South Korean undergraduate engineering students.
3. Comment 3
The abstract could use a sentence about included databases.
Response: We thank the reviewer for pointing this out.
Revision: We have added a sentence in the Abstract specifying the databases searched (e.g., RISS, KCI, and DBpia).
- Comment 4
On page 5 there is something wrong with the numbering. The numbering should start one row lower than it does currently.
Response: We thank the reviewer for carefully noticing this formatting error.
Revision: The numbering on page 5 has been corrected so that it aligns properly with the text.
- Comment 5
There are some minor terminology errors that should be fixed (e.g., “affective-psychological” vs. “affective/psychological”).
Response: We thank the reviewer for the observation. We carefully reviewed the manuscript and found that all occurrences were revised to use the standardized form ‘Affective–Psychological.’ This form has been consistently applied across headings, tables, and text to ensure conceptual clarity and terminological consistency.
Revision: This form of ‘Affective–Psychological’ has been consistently applied across headings, tables, and text to ensure conceptual clarity and terminological consistency.
- Comment 6
Some of the tables’ captions are missing explanations of categories and denominators.
Response: We thank the reviewer for pointing out this issue. We agree that clearer captions and explicit references to categories and denominators would improve the readability of our tables. We have revised the captions of Tables 3, 4, and 5 to include the total number of studies (N) for each category. In addition, we clarified in the text the categories examined and the frequencies/percentages calculated out of the total. We believe these revisions ensure that both the captions and the text now provide sufficient explanations of categories and denominators.
Revision:
- Table 3 caption: Table 3. Key variables and analysis of studies on affective–psychological characteristics (N= 38).
- Table 4 caption: Table 4. Key variables and analysis of studies on social characteristics (N= 43).
- Table 5 caption: Table 5. Key variables and analysis of studies on integrated characteristics (N= 14).
- In Sections 4.2.1–4.2.3, we explicitly described the categories (e.g., self-efficacy, stress, personality, communication, leadership, etc.) and presented their frequencies and percentages out of the total N.
We would like to sincerely thank the reviewers once again for their careful reading, insightful comments, and constructive suggestions. Their feedback has been invaluable in improving the clarity, precision, and overall quality of our manuscript. We believe that the revisions made in response to the reviewers’ comments have strengthened the contribution of this study.

Round 2
Reviewer 3 Report
Comments and Suggestions for Authors
I congratulate the authors on a very interesting and relevant paper as well as a constructive and serious review process